# Phosphate Removal Mechanisms in Aqueous Solutions by Three Different Fe-Modified Biochars

**DOI:** 10.3390/ijerph20010326

**Published:** 2022-12-25

**Authors:** Yiyin Qin, Xinyi Wu, Qiqi Huang, Jingzi Beiyuan, Jin Wang, Juan Liu, Wenbing Yuan, Chengrong Nie, Hailong Wang

**Affiliations:** 1School of Environmental and Chemical Engineering, Foshan University, Foshan 528000, China; 2School of Food Science and Technology, Foshan University, Foshan 528000, China; 3Foshan Engineering and Technology Research Center for Contaminated Soil Remediation, School of Environmental and Chemical Engineering, Foshan University, Foshan 528000, China; 4School of Environmental Science and Engineering, Key Laboratory of Water Quality and Conservation in the Pearl River Delta, Ministry of Education, Guangzhou University, Guangzhou 510006, China

**Keywords:** adsorption, phosphorus, eutrophication, modified-biochar

## Abstract

Iron-modified biochar can be used as an environmentally friendly adsorbent to remove the phosphate in wastewater because of its low cost. In this study, Fe-containing materials, such as zero-valent iron (ZVI), goethite, and magnetite, were successfully loaded on biochar. The phosphate adsorption mechanisms of the three Fe-modified biochars were studied and compared. Different characterization methods, including scanning electron microscopy/energy-dispersive spectrometry (SEM-EDS), Fourier transform infrared spectroscopy (FTIR), and X-ray photoelectron spectroscopy (XPS), were used to study the physicochemical properties of the biochars. The dosage, adsorption time, pH, ionic strength, solution concentration of phosphate, and regeneration evaluations were carried out. Among the three Fe-modified biochars, biochar modified by goethite (GBC) is more suitable for phosphate removal in acidic conditions, especially when the pH = 2, while biochar modified by ZVI (ZBC) exhibits the fastest adsorption rate. The maximum phosphate adsorption capacities, calculated by the Langmuir–Freundlich isothermal model, are 19.66 mg g^−1^, 12.33 mg g^−1^, and 2.88 mg g^−1^ for ZBC, GBC, and CSBC (biochar modified by magnetite), respectively. However, ZBC has a poor capacity for reuse. The dominant mechanism for ZBC is surface precipitation, while for GBC and CSBC, the major mechanisms are ligand exchange and electrostatic attraction. The results of our study can enhance the understanding of phosphate removal mechanisms by Fe-modified biochar and can contribute to the application of Fe-modified biochar for phosphate removal in water.

## 1. Introduction

Phosphorus (P) is one of the essential nutrients for plants. It is also the dominant contributor to eutrophication, which damages the aquatic environment and water quality. Currently, large amounts of P-containing products are widely used in human activities, such as in washing powders, detergents, fertilizers, and pesticides, leading to increasing P levels in natural water bodies [1]. Municipal wastewater treatment plants can only partially remove P, leading to large amounts of P being discharged into the environment [2]. Additionally, runoff from agricultural lands on which P-containing fertilizers and pesticides have been extensively applied is another dominant cause of increasing P levels. A level as low as 0.02 mg L^−1^ dissolved P can cause eutrophication by initiating algae bloom and high consumption of dissolved oxygen, which pose adverse effects on the environment [3,4].

By contrast, insufficient plant-available P in the soil is frequently reported in many countries [5]. Some P-containing minerals, such as apatite, are the domain sources for commercial P, which are expected to be depleted during the next 50 to 100 years. However, the demand for P is steadily growing at a rate of 2.5% per year [6,7]. Notably, it is difficult to rapidly regenerate this resource within decades; thus, developing methods of reusing P in a sustainable way is an urgent issue [2]. Phosphate recovery in water bodies or wastewater is difficult, owing to its relatively low concentration and its high hydration energy (2773 kJ mol^−1^) [8]. Typical methods included ion exchange, chemical precipitation, coagulation/flocculation, electrocoagulation, biochemical degradation, biological retention, and the use of artificial wetlands; however, these are expensive to maintain and manage, and they produce large amounts of waste [6]. Adsorption is one of cost-effective technologies available to remove and reuse P from water, even at low concentrations [2,9]. Traditional adsorbents for removing P include natural minerals (e.g., iron oxides and aluminum oxides, ferric hydroxides, and calcium hydroxides), chitosan, etc. [10].

Biochars are carbon-rich adsorbents which are mainly derived from organic wastes, including agricultural solid wastes and municipal sludge, and they have gained intensive research attention in the past decade [11,12,13]. Biochar is frequently used in environmental studies to adsorb and/or immobilize contaminants in both water and soil [[[14],[[15],[[16],[17]. Nevertheless, the adsorption capacity of pristine biochars in regards to P or other anionic contaminants is limited, compared their ability to adsorb other cationic potentially toxic elements, such as Pb and Cd. This characteristic is highly associated with the electrostatic repulsion caused by the negative surface charge of biochar and the anionic features of phosphates, in most cases [1,18,19]. Yao et al. [20] studied P removal by biochars produced by various feedstocks at 400 to 600 °C, including sugarcane bagasse, peanut hull, Brazilian pepperwood, and bamboo, respectively. Most of the biochars showed negligible P removal, with the highest removal efficiency of 3.1%. However, previous studies showed that Fe oxides enhance the removal of phosphate because of the large amounts of hydroxyl radicals on the surface, as well as the fact that Fe leaching can precipitate phosphate [21]. Herein, some studies have tried embedding different Fe oxides on the biochar surface to increase phosphate removal. For example, Zhu et al. [22] successfully loaded α-Fe_2_O_3_/Fe_3_O_4_ on bamboo biochars to remove PO_4_^3−^, whose maximum adsorption capacity (Q_m_) of PO_4_^3−^ was 2.81 mg g^−1^, according to the isothermal adsorption experiment at 45 °C.

In this study, three different Fe-containing biochars were selected to identify the key mechanisms for P removal and the regeneration of wastewater. Briefly, in this study: (1) the adsorption capacity of different Fe-modified biochars to PO_4_^3−^ was elaborated by batch experiments; (2) different kinetics and isotherm models were established to study the adsorption process of PO_4_^3−^; and (3) the mechanisms of PO_4_^3−^ adsorption and regeneration abilities of various biochars were thoroughly studied. Our work can contribute to a better understanding of the phosphate removal mechanisms of various Fe-modified biochars and their practical applications.

## 2. Materials and methods

### 2.1. Production of Pristine and Fe-Modified Biochars

All the solutions in the experiment were prepared using deionized water (DIW, 18 MΩ) and chemicals at analytical grade, without further purification. In the current study, lychee twig was used as the feedstock to prepare pristine and three Fe-modified biochars. The feedstock was collected in the countryside near Yangjiang City, Guangdong Province, China. It was rinsed by DIW and oven-dried at 80 °C for 48 h until reaching a constant weight. The dried feedstock was powdered by a grinder and then screened < 2 mm. An extract sieving of the feedstock was conducted for ZVI-modified biochar, which requires a particle size below 0.15 mm. The pristine biochar (BC) was pyrolyzed at 600 °C at a heating rate of 5 °C min^−1^, with flushing of N_2_ (0.3 L min^−1^) for 1 h in a tube furnace (Shengli SLG1100, Shanghai, China). Three different types of Fe-containing modified biochars, with different mechanisms, were produced, and their detailed pyrolysis procedure is described as follows.

FeSO_4_-modified biochar (CSBC) was prepared using two-step pyrolysis by mixing the feedstock with 1.0 mol L^−1^ FeCl_3_ and 72% H_2_SO_4_ at a ratio of 1:10:5 (g:mL:mL) in a crucible in an ultrasonic bath for 2 h [23,24]. The mixture was then incubated at 60 °C for 12 h and filtered with filter papers. The solid residue was carefully collected and then dried at 80 °C for 24 h for a consistent weight. Subsequently, the treated feedstock was pyrolyzed under the protection of N_2_ (0.3 L min^−1^), with a ramping rate of 10 °C at 600 °C for 2 h. Lastly, the CSBC was ground and sieved to <0.18 mm.

Goethite-modified biochar (GBC) was prepared by mixing 100 mL KOH (5 mol L^−1^) and 50 mL Fe(NO_3_)_3_ (1.0 mol L^−1^) with 3.2 g BC for 1 h, and then it was heated for 60 h at 70 °C [25,26]. The obtained raw modified biochar was washed with DIW and dried at 60 °C for 24 h. The dried biochar was ground and then passed through a 2 mm sieve again before storage.

ZVI-modified biochar (ZBC) was produced by mixing 1.93 g FeCl_3_∙6H_2_O and 20 g of feedstock with 800 mL of DIW for 24 h, followed by heating at 80 °C in a water bath for 8 h, using a method adapted from that of Han et al. [27]. The treated feedstock was separated by filtration with filter papers and dried at 80 °C for 12 h. The obtained particles were ground and passed through a 0.15 mm sieve before a secondary pyrolysis procedure was conducted at 900 °C for 2 h with N_2_ (0.4 L min^−1^).

### 2.2. Characterization of Pristine and Fe-Modified Biochars

The pH of each biochar was measured by mixing the biochar with DI water at a solid-to-liquid ratio of 1:20 (g mL^−1^) for 1 h at 30 rpm in an end-over-end rotator [28]. Brunauer–Emmett–Teller (BET) and Barret–Joyner–Halenda (BJH) sorption-desorption methods were used to analyze the specific surface area and pore characteristics of the pristine and modified biochars using N_2_ at 77.3 K with a surface area and pore size analyzer (Beishide BSD-PS, Beijing, China). The surface morphology of the biochars was studied by a scanning electron microscope (SEM, FEI S400, Hillsboro, OR, USA) equipped with an energy dispersive spectrometer (EDS), except for CSBC, which was studied by another SEM (Phenom-World Prox, Eindhoven, Netherlands)owing to its magnetic properties. An elemental analyzer (Elementar UNICUBE, Hanau, Germany) was used to determine the total content of C/H/N/O/S of the biochars.

To better understand the mechanisms of PO_4_^3−^adsorption on modified biochar, the variations in the surface functional groups and element morphology of the biochar samples after adsorption were analyzed via Fourier transform infrared spectroscopy (FTIR, Shimadzu IRAffinity-1S, Kyoto, Japan) and X-ray photoelectron spectroscopy (XPS, Thermo Scientific ESCALAB 250Xi, USA). The spectra of FTIR were collected in the scanning range of 400–4000 cm^−1^ at a resolution of 4 cm^−1^. XPS analysis was performed using a monochromatized Al Ka X-ray source (1486.6 eV). The adsorption experiments for FTIR and XPS were conducted with an increased initial concentration. Specifically, 0.1 g biochar was mixed with 20 mL PO_4_^3−^ solution (0.1 g L^−1^ P) for 24 h at pH 6. The biochar was separated using filter papers and oven-dried at 80°C to a constant weight.

The pH of the biochar was measured by mixing the biochar with DI water at a solid-to-liquid ratio of 1:20 (g mL^−1^) for 1 h at 30 rpm in an end-over-end rotator [28]. The total Fe content of the pristine and modified biochars was analyzed by inductively coupled plasma optical emission spectroscopy (ICP-OES, PerkinElmer Avoi 200, Waltham, MA, USA), and subjected to acid digestion by HNO_3_ (68% *w*/*w*) and HCl (37% *w*/*w*) at a volumetric ratio of 2:1 in a microwave digester (CEM MARS6, Matthews, NC, USA). The detailed digestion procedure was documented in our previous work [24].

### 2.3. Batch Adsorption Experiments

The PO_4_^3−^ stock solution (1.5 g L^−1^ P) was prepared using potassium dihydrogen phosphate (KH_2_PO_4_). Batch adsorption experiments were conducted to evaluate adsorption kinetics, adsorption isotherms, and the effects of pH, ion strength, and dosage of the PO_4_^3−^ adsorption on the pristine and modified biochars. Without specific clarification, the adsorption batch experiments were performed at a biochar-to-solution ratio of 1:200 (g:mL), 30 rpm, and at room temperature (25 ± 1 °C) by an end-over-end rotator for 4 h. The PO_4_^3−^ working solution (15 mg L^−1^ P) was adjusted to pH = 6.0 (by 0.1 M HCl or 0.1 M NaOH, if necessary), with 0.01 M NaNO_3_ as the background electrolyte. After mixing, the samples were filtered by mixed cellulose ester (MCE) membrane filters, and the PO_4_^3−^ concentrations were determined, as described in Section 2.5.

The adsorption kinetics study of PO_4_^3−^ was conducted for 5 to 1440 min for BC, GBC, ZBC, and CSBC. Pseudo-first-order, pseudo-second-order, and intra-particle diffusion models were applied. Adsorption isotherms were studied by adsorption batch experiments using various initial concentrations of PO_4_^3−^ (5 to 200 mg L^−1^ P) for adsorption at equilibrium, and the Langmuir, Freundlich, and Langmuir–Freundlich isotherm models were used to describe the adsorption characteristics. The detailed calculations and fitting parameters of both the adsorption kinetics and the isotherms are documented in the Appendix A. The effects of biochar dosage, pH, and anion strength of the adsorption solution were investigated as well. Specifically, the adsorption batch experiment was performed using 0.1 g pristine or modified biochar and the PO_4_^3−^ working solution for 4 h at a biochar-to-solution ratio of 1:200 g mL^−1^. The effects of solution pH were investigated at pH 2–10. After comparing the pH, the optimal adsorbent dosage was studied in the range of 0.5–6 g L^−1^. The effects of the anion strength of NO^3−^ of 0.005 M, 0.01 M, and 0.05 M NO_3_^−^ were compared, respectively, and the effects of the anion strength of PO_4_^3−^ on the adsorption of biochar were evaluated.

### 2.4. Desorption and Regeneration

To evaluate the reuse capacity of the three Fe-modified biochars, desorption and regeneration experiments were conducted. Adsorption experiments were similarly performed, as described above. Desorption experiments were conducted with a subsequent washing by 0.1 M NaCl solution at a biochar-to-solution ratio of 1:200 (g mL^−1^) at 30 rpm in an end-over-end rotator for 24 h. The desorption rate was calculated by the following formula [29]:$$\mathrm{Desorption}\;\mathrm{rate}=\frac{\left(C_{l}\times V_{2}\right)}{M_{2}\times Q}\times 100\%$$
where C_l_ is the concentration of PO_4_^3−^ after desorption experiments, mg L^−1^, V_2_ is the volume of the desorption solution (mL), and M_2_ (g) is the mass of the biochar samples in the desorption experiments.

Regeneration experiments were conducted by washing the samples with 0.5 M NaOH solution after the adsorption of PO_4_^3−^. The regenerated biochar was rinsed with DIW, oven-dried, and then mixed again with 20 mL PO_4_^3−^ (15 mg L^−1^ P) solution. Three successive adsorption/desorption cycles were performed, and the concentration of PO_4_^3−^ in the solution of the regenerated biochars in each cycle was determined.

### 2.5. Determination of P and Statistical Analysis

After the above batch experiments were completed, all the samples were filtered using 0.45-μm MCE membrane filters and then stored at 4 °C for P concentration. The concentration of P in the PO_4_^3−^ working solution and samples was analyzed based on the Bray-1 method using a UV-vis spectrometer at 882 nm (UV-Vis, Yidian752N Plus, Shanghai, China). The detailed experimental steps of the analysis of P are documented in the Appendix A. All the adsorption experiments were carried out in triplicate. The results are expressed as averages and are presented with standard deviations in the figures.

## 3. Results and Discussion

### 3.1. Characterization of Pristine and Fe-Modified Biochars

After modification, the Fe contents of all the modified biochars were remarkably increased, with an order of CSBC (235.6 g kg^−1^) > GBC (165.6 g kg^−1^) > ZBC (15.62 g kg^−1^) (Table 1), indicating a successful loading of Fe. It can be further supported by the SEM-EDS results that Fe peaks were found (Figure 1d,f,h). Impressively, ZBC showed the lowest total Fe content among the three Fe-modified biochars, and this result is strongly associated with a smaller loading in its modification. The pH of the biochar is also significantly changed after modification (Table 1). The pH of pristine biochar is 10.14, while the pH of CSBC sharply decreased to 3.30 because of the use of H_2_SO_4_. The pH of GBC slightly reduced to 9.54, while interestingly, the pH of ZBC increased to 11.97.

Generally, high pyrolytic temperatures lead to a high surface area of the biochar [30]. Among the four different biochars, ZBC has the highest pyrolytic temperature of 900 °C, although it has the highest surface area (261.7 mg g^−1^) (Table 1), which is possibly associated with the ZVI itself. GBC and CSBC were produced at the same temperature, while GBC has the lowest surface area of 6.13 mg g^−1^, which might be associated with the mineral aggregation of the goethite formed (Figure 1c). The heterogeneously formed minerals on the surface of GBC can block the surface pores and reduce the surface area [31]. However, the formed minerals might lead to a higher adsorption of P because they can complex with phosphate. CSBC showed well-developed porous structures with fewer impurities, which could be due to the use of acidic washing during the modification process [24,32,33].

### 3.2. Effect of Solution pH on the P Adsorption

Phosphate speciation is strongly affected by the solution pH, in addition to the adsorbent surface. Phosphate exists in solution as H_3_PO_4_, H_2_PO_4_^−^, HPO_4_^2−^, and PO_4_^3−^ at different ratios, which is strongly associated with the solution pH. At pH 3–11, H_2_PO_4_^−^ and HPO_4_^2−^ are mainly found, while H_2_PO_4_^−^ has a higher adsorption capacity [34,35]. H_2_PO_4_^−^ is the dominant phosphate when pH < 5, while HPO_4_^2−^ and PO_4_^3−^ are the dominant species in a pH range of 5–12 [36,37]. In the current study, the total phosphate as a sum of the above P-containing anions was determined by the Bray-1 method.

The removal efficiencies of PO_4_^3−^ by various biochars were significantly affected by pH variation (Figure 2). Impressively, the PO_4_^3−^ removal by ZBC reached nearly 87.52–99.54% at pH 3–10; however, it was remarkably hindered at pH 2. This could be due to the fact that part of the Fe oxides formed on the ZBC surface were dissolved due to the low pH. Our ICP-OES results can further support this hypothesis. Dissolved Fe was detected in the solution after adsorption by ZBC (Appendix A), in amounts of 4.58 ± 0.09 and 30.84 ± 1.54 mg L^−1^ for pH 2 and pH 1, respectively. The amount of dissolved Fe was undetectable for pH 3 and pH 4 after adsorption by ZBC. Interestingly, the ICP-OES results also suggested that the Fe-containing minerals on the surface of GBC and CSBC can be dissolved by acidic conditions, starting from different pH levels. For CSBC, the pH is 3, and for GBC, the pH is 1. However, the dissolved amount of Fe for GBC is much higher (12 and 20 times, respectively) than for CSBC and ZBC at pH 1, owing to the fact that we used significantly larger amounts of Fe in the modification of GBC.

GBC showed an excellent removal efficiency of PO_4_^3−^ at pH 2, which is consistent with the results indicating that no dissolved Fe was determined by ICP-OES at pH 2 (Appendix A). This suggested that, in the current study, the Fe-containing minerals of GBC might be the most important components for PO_4_^3−^ adsorption [38]. Nevertheless, the adsorption capacity of P obviously decreased with the increase in the solution pH in the batch experiments.

CSBC showed the lowest removal of PO_4_^3−^ among the three modified biochars, in a range of 11.29–22.01% at pH 2–10. Notably, the dissolving of the Fe-containing minerals on the biochar surface was obvious starting from pH 3. However, higher amounts of dissolved Fe in the solution showed limited effects on the removal of phosphate, which might be associated with the low removal rate of phosphate by CSBC.

BC showed extremely low P adsorption, which is probably due to its negatively charged surface (Figure 2). Interestingly, BC can release 0.83–0.97 mg L^−1^ of phosphate, possibly from lignin of lychee branches, at pH 2–10 after extraction using DI water, with 0.01 M NaNO_3_ as the background electrolyte (Figure 3). The release of PO_4_^3−^ (1.68–3.63 mg L^−1^) was enhanced by the extractants with a lower pH (pH 2–3). The release of phosphate from the biochars was frequently observed, especially for those with high phosphate levels in the feedstocks [31,39]. For example, a range of 0.005 to 2.06 mg g^−1^ P was determined at pH 3–11 for sludge biochar produced at 400–700 °C [40]. These results also suggest that the lower pH of the leachate, the higher the amounts of P that will be released. Therefore, solution pH = 7 was selected in this study for the following batch experiments.

### 3.3. Adsorption Kinetics and Isotherm

The adsorption of P by both ZBC and GBC can be divided into a fast and a slow adsorption stage, which quickly reached adsorption equilibrium in 120 and 240 min, respectively, compared with CSBC, which exhibits steady adsorption (Figure 4a). The adsorption equilibrium of CSBC on PO_4_^3−^ was reached after 10 days, with a final removal of 97.47% (Appendix A). Considering the cost-effectiveness and the better comparison qualities, we consistently used 4 h as the time period in the batch experiments. Our results suggested that the pseudo-second-order model might be more fitting to describe the adsorption behaviors of the three Fe-modified biochars, compared with the pseudo-first-order model, indicating that chemisorption occurred in the removal of P [31]. More fitting parameters can be found in Table 2.

The three-stage intra-particle diffusion model was used to further analyze the adsorption processes of P on the Fe-modified biochars (Figure 4b). GBC and ZBC exhibit slippery slopes in the first stage (within 1 h), indicating that the GBC and ZBC rapidly removed PO_4_^3−^ by the adsorption sites provided by the Fe ions in the first stage; then, the adsorption sites were gradually occupied. Remarkably, the easily accessible sorption sites of ZBC were quickly occupied, and the adsorption reduced distinctly. The adsorption by CSBC showed a significant difference from ZBC and GBC; a slow adsorption within the first 1 h and a long and flat adsorption in the second and third stages were observed.

The adsorption results of PO_4_^3−^ by ZBC and GBC agreed well with both the Langmuir and Freundlich models, with high correlation coefficients (R^2^) of over 0.9 (Figure 5 and Table 3). The adsorption results of CSBC, GBC, and ZBC fitted better with the Freundlich model, owing to the higher R^2^ values of over 0.95, indicating that the adsorption more likely occurred on the heterogeneous surface, which tended toward a multi-layer adsorption [41]. This result also agreed with the results of the kinetics studies in which Fe-modified biochar fitted better with the pseudo-second-order double-layer adsorption. Some other previous studies also suggested similar results, determining that the adsorption of P by modified biochars fitted better with the Freundlich model [42].

The Q_m_ (maximum sorption capacities) obtained by calculation via the Langmuir model are 0.005, 12.63, and 8.254 mg g^−1^ for CSBC, GBC, and ZBC, respectively (Table 3). For the Langmuir–Freundlich model, the calculated Q_m_ are 2.88, 12.33, and 19.66 mg g^−1^ for CSBC, GBC, and ZBC, respectively. Although the adsorption capacity of the Fe-modified biochar is relatively low compared with some other adsorbents (Appendix A), for example, biochar modified by Mg or Ca [18,43], it is significantly improved compared with the pristine biochar. Compared with some Fe-modified biochars, such as Fe-impregnated biochars derived from wood chip (Q_m_ = 3.201 mg g^−1^), and magnetic biochars derived from water hyacinth (Q_m_ = 5.07 mg g^−1^), in the current study, GBC and ZBC exhibit better adsorption effects. The poor affinity of the pristine and modified biochars to P is probably owing to its negatively charged surface. However, it should be noted that Q_m_ is the theoretical maximum adsorption capacity, which can be different from the experimental maximum adsorption.

### 3.4. Effect of Biochar Dosage and Ionic Strength

Notably, a higher biochar dosage and a higher removal efficiency of PO_4_^3−^ was found after an investigation using the biochar dosage from 0.5 to 6 g L^−1^ in the adsorption batch experiments (Figure 6). The removal efficiencies of PO_4_^3−^ increased greatly between 1–2, 4–5, and 2–3 g L^−1^ for ZBC, GBC, and CSBC, respectively. The adsorption capacity became saturated when the dosage of ZBC reached 3 g L^−1^. However, owing to the removal efficiency and cost-effectiveness of all three biochars, 5 g L^−1^ was finally selected in our study.

The ionic strength of 0.005 to 0.05 M in the solution was evaluated for the PO_4_^3−^ adsorption by CSBC, ZBC, and GBC (Figure 7). The coexistence of NO_3_^−^ only slightly affected the adsorption of P by CSBC and GBC. Specifically, it enhanced the PO_4_^3−^ adsorption for ZBC. This might have resulted from the additional NO_3_^−^ enhancing the oxidation of Fe^0^ to Fe^II^, possibly leading to the release of Fe^2+^ ions and the formation of precipitants with PO_4_^3−^ [44]. A higher dosage of the background electrolyte (NO_3_^−^) as 0.05 M can only marginally enhance the removal of PO_4_^3−^.

### 3.5. Desorption and Regeneration

The desorption experiments were conducted by NaCl washing. CSBC and GBC showed higher desorption efficiencies for PO_4_^3−^, which are 16.56% and 11.32%, respectively, in the first desorption cycle, compared with ZBC (Figure 8). With the increase in the desorption cycle times, CSBC and GBC showed low desorption efficiency (below 0.92%) in the 2nd and 3rd desorption cycles. However, the desorption efficiency of PO_4_^3−^ by ZBC gradually increased with the washing times at the rates of 3.53%, 6.09%, and 7.96%, respectively. This might indicate that the mechanisms of PO_4_^3−^adsorption by ZVI are significantly different from those of CSBC and GBC, which will be discussed in detail in Section 3.6. After washing three times with NaCl, the total desorption efficiency of GBC was less than 12%, and it can be considered the most reliable material among the three studied Fe-modified biochars.

Regeneration experiments for the Fe-modified biochars were also conducted in triplicate. The results showed that although ZBC showed the highest removal efficiency of PO_4_^3−^ (96.52%) for the first washing, the capacity of PO_4_^3−^ removal by ZBC reduced sharply (24.98%) after desorption by NaOH (Figure 9). The removal efficiency by ZBC was further decreased by the second washing by NaOH to 12.55%. This is strongly related to the ZVI aging effect and deactivation, and some previous studies have reported similar results [45,46].

GBC and CSBC showed better regeneration capacities; however, the removal capacity of PO_4_^3−^ by CSBC was poor (Figure 9). After washing with NaOH, the removal efficiency by GBC only marginally decreased from 68.62% to 67.66%. After three successive cycles, the removal efficiency of PO_4_^3−^ was reduced to 52.93%, indicating a moderate reusability. This could be owing to the fact that some attractive sites on the GBC could not be fully recovered as due to the phosphate tightly attached to the surface [47]. The adsorption efficiency of the second and third cycles were 98.60% and 77.13% of those of the first cycle, respectively.

### 3.6. Possible Mechanisms of Fe-Modified biochar for P Adsorption

The FTIR and XPS analyses of Fe-modified biochars, before and after P adsorption, are presented in Figure 10 and Figure 11, respectively. A special P adsorption with a higher initial P concentration was completed. The full range of XPS spectra (Appendix A) and the spectra of the P 2p binding energies for all Fe-modified biochars indicated the successful adsorption of P (Figure 11).

The peaks near 3200 and 3439 cm^−1^ are due to the stretching vibration of the -OH bond, which is associated with the Fe hydroxyl on the Fe oxides surface (Figure 10). Moreover, the bands near 1100 cm^−1^ occur due to the stretching vibration of the P-O bond [48]. The strengthening peaks of the P-O bonds and the disappearance of the -OH bond after the P adsorption indicate the ligand exchange via the adsorption replacing the hydroxyl groups, which are more obvious for GBC in this study. The results of XPS supported that FeOOH (goethite), which has a good affinity to PO_4_^3−^, was successfully loaded on GBC before P adsorption, with a binding energy of 727.24 eV [21]. The weakening of the peaks of the Fe 2p spectrum of the adsorbed GBC also confirmed that FeOOH plays an important role in the adsorption process of PO_4_^3−^ (Figure 11i) [46,47].

Previous studies suggested that different Fe oxides have various P adsorption capacities. Zhang et al. [21] produced coconut-derived biochars at 900 °C and impregnated them with various Fe minerals using co-precipitation methods. Their results suggested that coconut-derived biochars embedded with goethite reached the highest maximum adsorption capacity (22.14 mg g^−1^), while biochar embedded with magnetite exhibited the lowest (9.408 mg g^−1^) capacity. This result is consistent with our findings that GBC, which mainly contains goethite due to its production method6, has a higher P removal capacity, while CSBC exhibits low P removal, which might be related to the fact that the formed minerals are mainly magnetite (Fe_3_O_4_) and natrojarosite (NaFe_3_(SO_4_)_2_(OH)_6_), according to X-ray diffraction analysis [23]. The suggested major P removal mechanisms for the biochar impregnated with Fe oxides are electrostatic attraction and ligand exchange (especially for the Fe hydroxyl). Zhang et al. [21] also suggested that impregnation with goethite can increase the pH_zpc_ of biochars, compared with other Fe oxides and the pristine biochar, and it also enhances the electrostatic attraction.

Surface precipitation, such as the formation of FePO_4_, was proposed as one of the major P removal mechanisms by the Fe-modified biochars, as shown in the equations below [45]. Before adsorption, the Fe 2p spectrum of ZBC can be divided into Fe^0^, Fe 2p_3/2_, and Fe 2p_1/2_, and the Fe 2p_3/2_ are related to iron Fe^2+^(711.44 eV) and Fe^3+^(714.06) [49]. However, owing to the low Fe content of ZBC, the XPS analysis after P adsorption is difficult to analyze in the current study. Ai et al. [45] found a decline in the intensity of Fe^2+^ with an increase in Fe^3+^ peaks, suggesting that the oxidation and precipitation of FePO_4_ might occur. This is consistent with the desorption and recycling experiment results. The active sites that have formed precipitates cannot be recovered by washing with 0.1 M NaOH in the recycling experiment. Although ZBC has the highest removal efficiency, it can hardly be recovered as an absorbent. Moreover, the formed precipitates can be dissolved again, remarkably disturbing the removal efficiency of P, once the solution pH reaches 2. The P removal was enhanced by the increase in pH (8–10) for ZBC, owing to the fact that alkaline conditions are preferred for precipitate formation.
2*Fe*^0^ +*O*_2_ +2*H*_2_*O*→2*Fe*^2+^ +4*OH^−^*(1)
*Fe*^0^ +2*H*_2_*O* →*Fe*^2+^ +*H*_2_↑ + 2*OH^−^*(2)
4*Fe*^2+^ +*O*_2_ +2*H*_2_*O*→4*Fe*^3+^ +4*OH^−^*(3)
*Fe*^3+^ +*PO_4_*^3*−*^→*FePO*_4_(4)

In the current study, the possible common removal mechanisms for P removal by the Fe-modified biochars are electrostatic attraction, ternary adsorption, ligand exchange (by replacing hydroxyl groups), surface complexation, and surface precipitation [2,21,38,45]. In the current study, for ZBC, the major mechanism of P removal is surface precipitation, while for GBC and CSBC, the major mechanisms are ligand exchange and electrostatic attraction.

## 4. Conclusions

Fe-modified biochars can act as suitable adsorbents for the removal of various contaminants, as they can be easily separated. In this study, various Fe-modified biochars were produced their effects on P removal were evaluated. P removal was greatly enhanced by the impregnation of Fe oxides and ZVI, compared with the results of BC. The Qm results, obtained via the Langmuir–Freundlich isothermal model, are 19.66 mg g^−1^, 12.33 mg g^−1^, and 2.88 mg g^−1^, for ZBC, GBC, and CSBC, respectively. ZBC showed a higher removal of P under pH 3–10; however, it has a poor desorption and regeneration capacity. The XPS and FTIR studies suggested that the removal mechanisms for the Fe-modified biochars are different. The dominant mechanism for ZBC is surface precipitation, while for GBC and CSBC, the major mechanisms are ligand exchange and electrostatic attraction. Goethite is more stable under acidic conditions; thus, GBC obtained the highest P removal at pH 2. However, it showed decreased P removal at pH 8–10, owing to the electronic repulsion. Although ZBC has the highest removal efficiency among the studied Fe-modified biochars, it might not be suitable for practical applications, as it is difficult to be reused.

## Figures and Tables

**Figure 1 ijerph-20-00326-f001:**
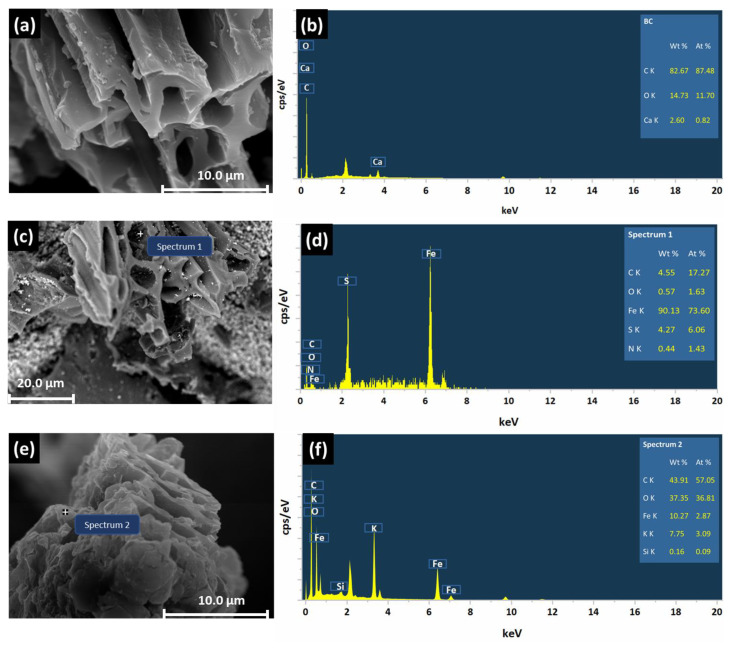
SEM-EDS images of BC, GBC, CSBC, and ZBC (**a**–**h**).

**Figure 2 ijerph-20-00326-f002:**
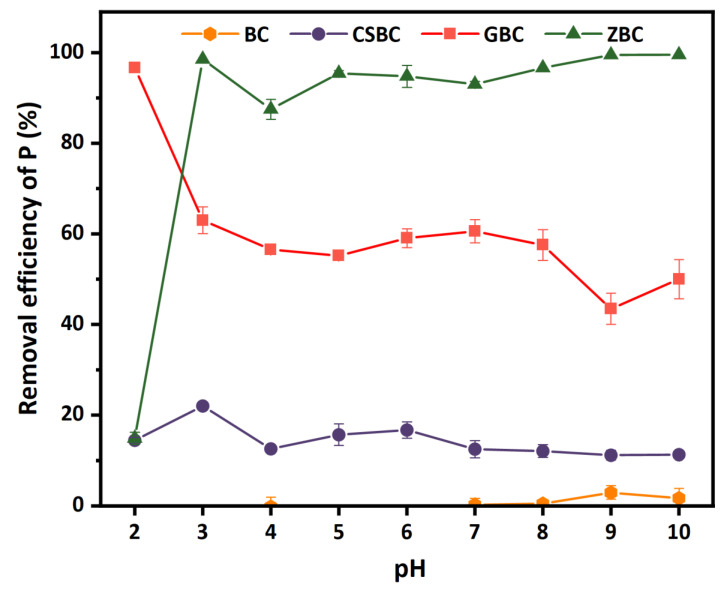
Effects of different pH levels on the P adsorption in BC, CSBC, GBC, and ZBC. (Adsorbent dose: 5 g L^−1^; P initial concentration: 15 mg L^−1^; adsorption time: 4 h; temperature: 25 °C; ionic strength: 0.01 M.)

**Figure 3 ijerph-20-00326-f003:**
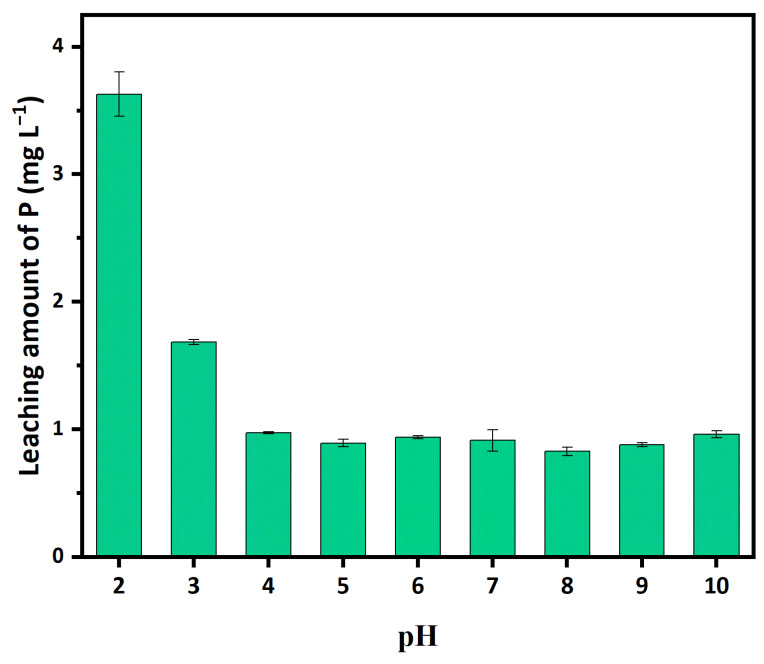
Effects of different pH on the leaching amount of P in BC. (Adsorbent dose: 5 g L^−1^; P initial concentration: 15 mg L^−1^; adsorption time: 4 h; temperature: 25 °C; ionic strength: 0.01 M.)

**Figure 4 ijerph-20-00326-f004:**
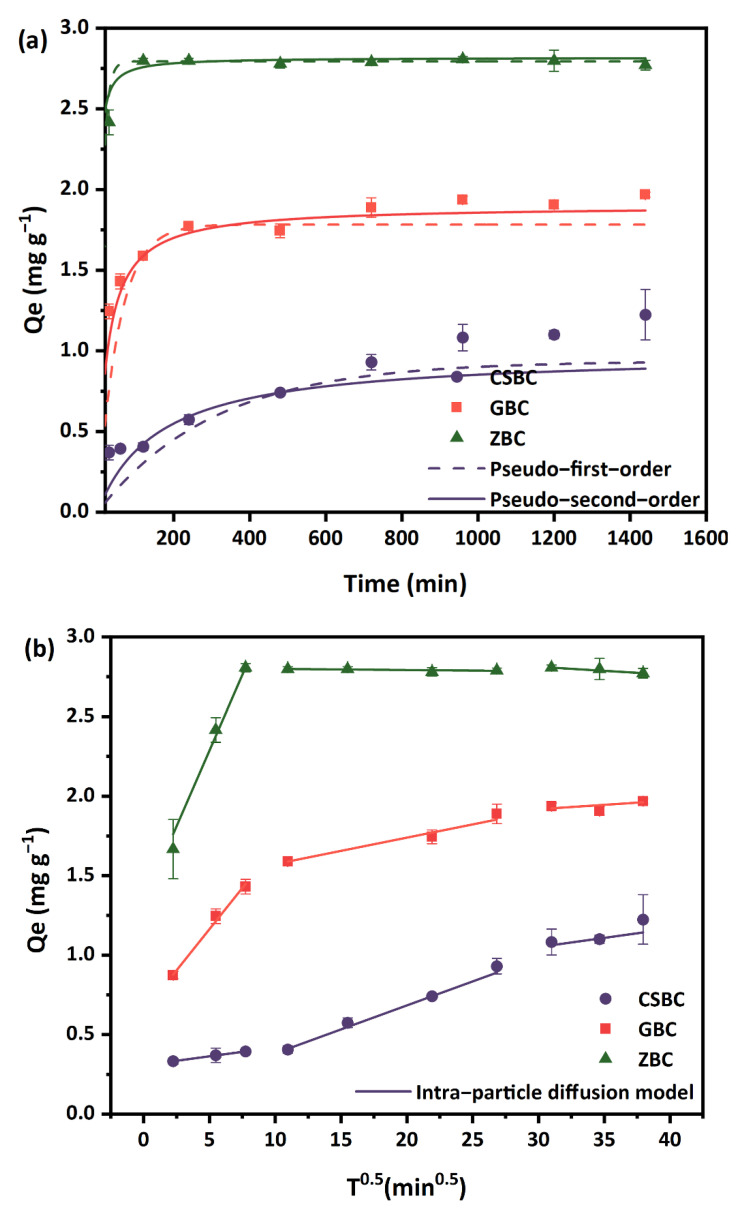
Adsorption kinetics of P by CSBC, GBC, and ZBC: (**a**) pseudo−first−order and pseudo−second−order models; (**b**) intra−particle diffusion model. (Adsorbent dose: 5 g L^−1^; P initial concentration: 15 mg L^−1^; pH: 6; temperature: 25 °C; ionic strength: 0.01 M.)

**Figure 5 ijerph-20-00326-f005:**
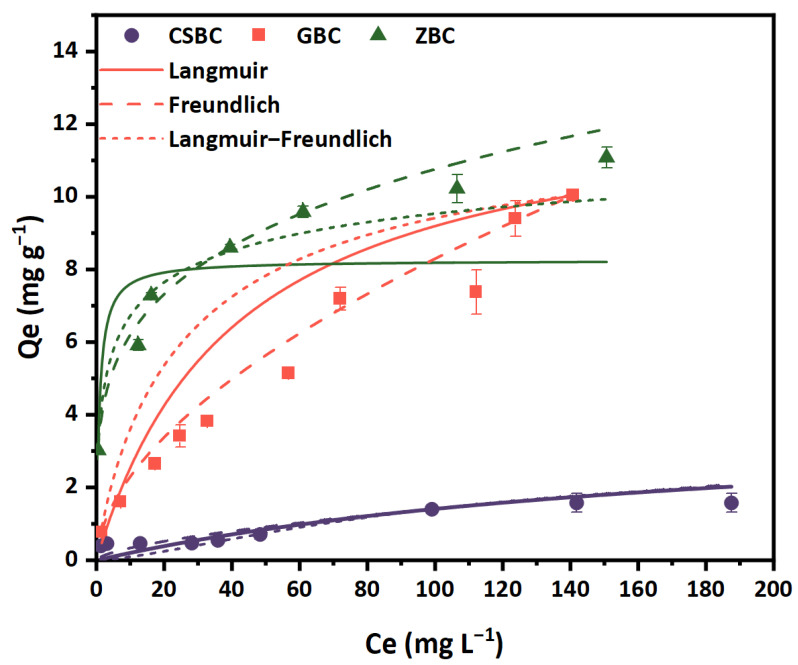
Adsorption isotherms of P by CSBC, GBC, and ZBC. (Adsorbent dose: 5 g L^−1^; adsorption time: 4 h (CSBC and GBC) and 8 h (ZBC); pH: 6; temperature: 25 °C; ionic strength: 0.01 M.)

**Figure 6 ijerph-20-00326-f006:**
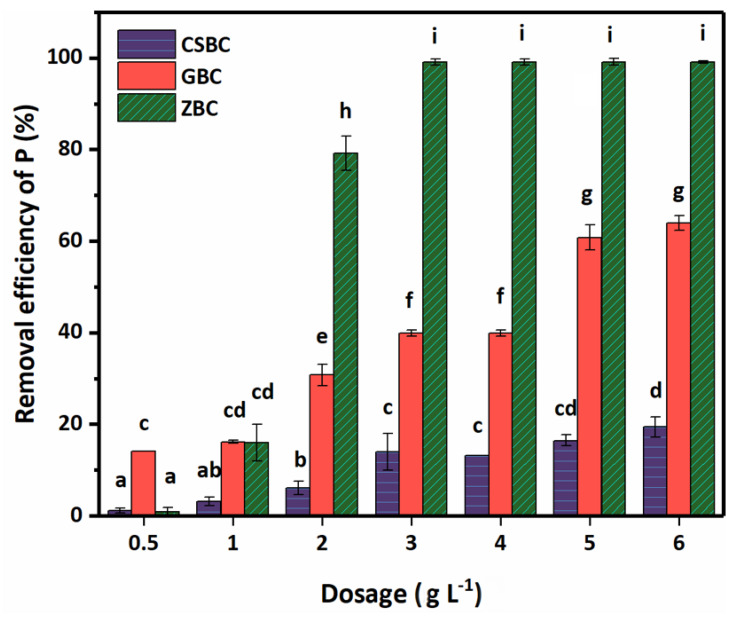
Effects of different dosages on the P adsorption by CSBC, GBC, and ZBC. (P initial concentration: 15 mg L^−1^; adsorption time: 4 h; pH: 6; temperature: 25 °C; ionic strength: 0.01 M.)

**Figure 7 ijerph-20-00326-f007:**
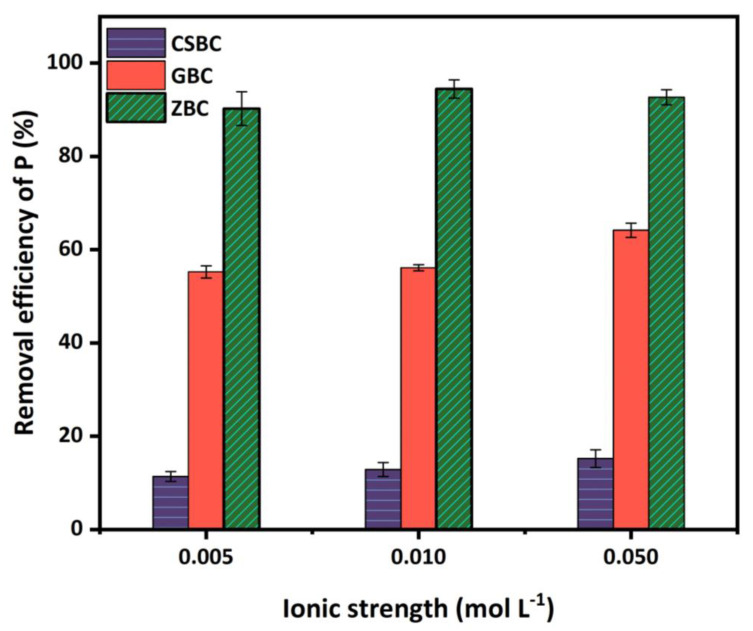
Effects of ionic strength on the P adsorption by CSBC, GBC, and ZBC. (Adsorbent dose: 5 g L^−1^; P initial concentration: 15 mg L^−1^; adsorption time: 4 h; pH: 6; temperature: 25 °C.)

**Figure 8 ijerph-20-00326-f008:**
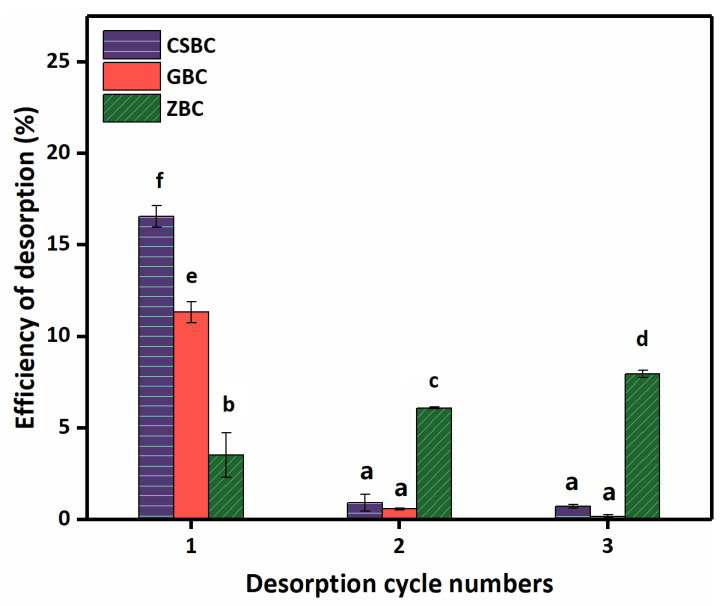
Effects of desorption of P by CSBC, GBC, and ZBC. (Adsorbent dose: 5 g L^−1^; P initial concentration: 15 mg L^−1^; adsorption time: 24 h; pH: 6; temperature: 25 °C; ionic strength: 0.01 M; desorption concentration: 0.1 M NaCl; desorption time: 24 h.)

**Figure 9 ijerph-20-00326-f009:**
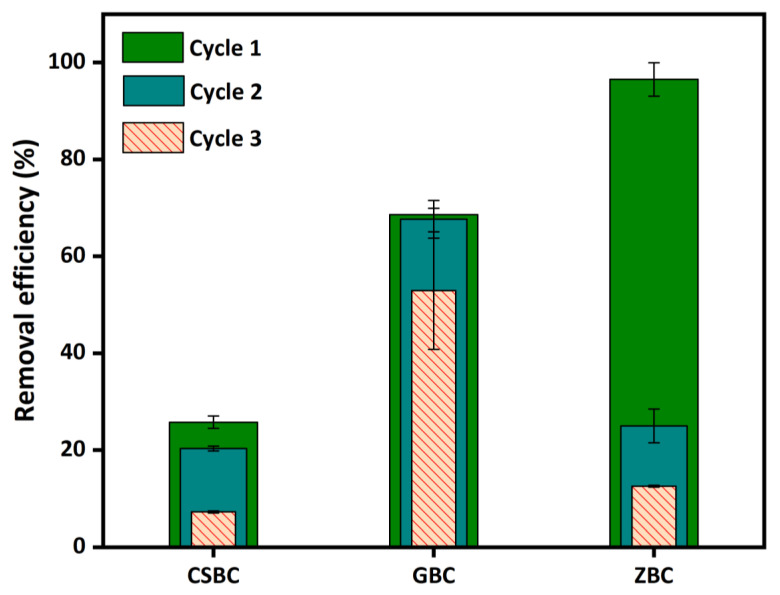
The removal efficiencies for P at each reuse cycle by the CSBC, GBC, and ZBC. (Adsorbent dose: 5 g L^−1^; P initial concentration: 15 mg L^−1^; adsorption time: 24 h; pH: 6; temperature: 25 °C; ionic strength: 0.01 M; desorption concentration: 0.5 M NaOH; desorption time: 24 h.)

**Figure 10 ijerph-20-00326-f010:**
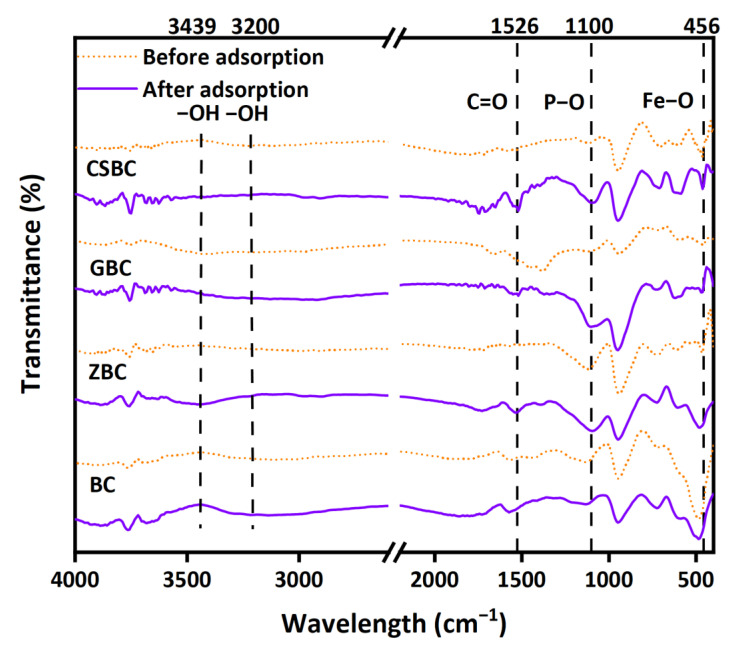
FTIR spectra of BC, CSBC, GBC, and ZBC, before and after P adsorption.

**Figure 11 ijerph-20-00326-f011:**
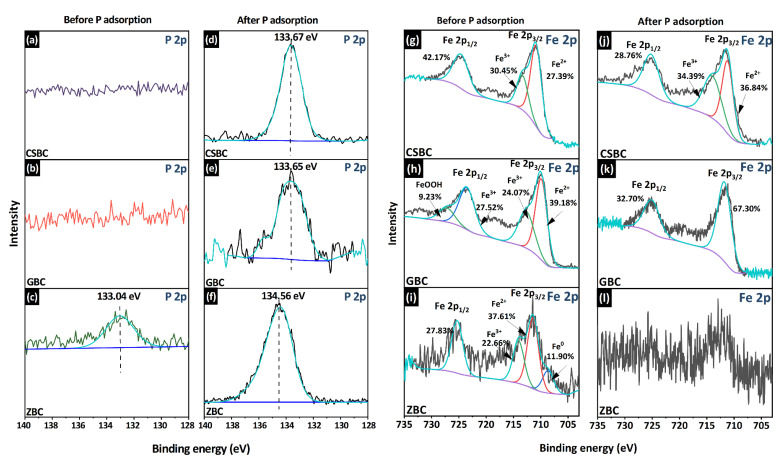
High-resolution XPS spectra of CSBC, GBC, and ZBC: (**a**–**c**) P 2p spectra before P adsorption; (**d**–**f**) P 2p spectra after P adsorption; (**g**–**i**) Fe 2p spectra before P adsorption; and (**j**–**l**) Fe 2p spectra after P adsorption.

**Table 1 ijerph-20-00326-t001:** Physicochemical properties of BC, GBC, CSBC, and ZBC.

Biochar	Physical Properties			Major Element					
	BET (m^2^ g^−1^)	Pore Volume (nm)	pH	C (%)	O (%)	H (%)	N (%)	S (%)	Total Fe Content (g kg^−1^)
BC	5.18	7.779	10.14	79.820	10.279	2.323	1.000	0.137	NA
CSBC	256.89	4.396	3.30	47.420	17.445	1.442	0.710	8.359	235.60
GBC	6.13	8.288	9.54	20.140	39.075	1.341	3.180	0.056	165.64
ZBC	261.70	5.193	11.97	87.450	5.984	0.744	0.470	0.102	15.62

**Table 2 ijerph-20-00326-t002:** Parameters of adsorption kinetics for P removal by GBC, CSBC, and ZBC.

	Pseudo-First Order			Pseudo-Second Order			Intra-Particle Diffusion								
Biochar	*Q_e_*	*K* _1_	*R* ^2^	*Q_e_*	*K_2_*	*R* ^2^	*C_1_*	*K_3_*	*R* ^2^	*C_2_*	*K* _4_	*R* ^2^	*C_3_*	*K* _5_	*R* ^2^
CSBC	0.935	0.003	0.587	1.223	0.006	0.685	0.307	0.011	0.999	0.080	0.030	0.994	0.704	0.012	0.463
GBC	1.783	0.018	0.607	1.90	0.022	0.796	0.645	0.104	0.995	1.406	0.016	0.981	1.753	0.006	0.455
ZBC	2.794	0.087	0.677	2.819	0.136	0.715	1.335	0.190	0.991	2.806	−7.082	0.596	2.961	−0.005	0.980

**Table 3 ijerph-20-00326-t003:** Parameters for the adsorption isotherm for P removal by GBC, CSBC, and ZBC.

	Langmuir			Freundlich			Langmuir–Freundlich			
Biochar	*Q_m_*	*K_l_*	*R* ^2^	*K_f_*	*n*	*R^2^*	*K_i_*	*n*	*Q_m_*	*R* ^2^
CSBC	0.005	4.023	0.901	0.082	1.621	0.950	0.001	1.44	2.88	0.848
GBC	12.63	0.003	0.989	0.637	1.794	0.999	0.053	0.892	12.33	0.981
ZBC	8.254	1.164	0.957	3.573	4.178	0.995	0.229	0.332	19.66	0.995

## Data Availability

All the research data have been included in the manuscript; others if any, are available from the corresponding author upon reasonable request.

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
