# Peer review of "Phosphate Removal Mechanisms in Aqueous Solutions by Three Different Fe-Modified Biochars"

_ijerph, 2022, doi:10.3390/ijerph20010326_

Round 1
Reviewer 1 Report
The author studied and compared the adsorption mechanism of phosphate on three kinds of Fe-modified biochar. The research work is systematic and comprehensive. The research results are helpful to enhance the understanding of removal mechanisms of phosphate by Fe-modified biochar, and contribute to the application of Fe-modified biochar on phosphate removal in water. However, the mechanism of P adsorption by three kinds of Fe-modified biochar has not been well proved by experiments, so I think more experiments should be added to be more convincing.
Introduction
1. Line 40: change “the other” to “another”.
2. Line 71: It is inappropriate to use “besides” here, it should be a transition relationship.
3. Line 83-85: The research significance needs to be rewritten. The focus of this paper should be to highlight the removal of phosphate.
Materials and methods
1. Line 103: “tried” ?, It should be “dried”.
Results and discussion
1. Line 207: It is inappropriate to use “yet” here, it should be a causal relationship.
2. Line 233: “starting from pH 3 for CSBC and pH 1 for GBC”. This sentence is contrary to the previous statement that the Fe-containing minerals can be dissolved under acidic conditions and needs to be rewritten.
3. Line 242: It is inappropriate to use “Besides” here, the conjunctions of the whole article need to be rechecked.
4. Line 248-250: This sentence can be deleted or rewritten because the previous study has no obvious significance for the present study.
5. Line 265: Please explain why the Elovich model was not used to fit the adsorption kinetics.
6. Line 266: Add “both” before "can".
7. Line 280-282: There needs to be a phenomenon description, such as the lines did not cross through the origin.
8. Line 289-293: These two sentences need to be supported by literature, and the last sentence is too long to express clearly.
9. Line 328-330: This sentence made no sense; I suggest deleting it.
10. Line 341: change “sharply reduced” to “reduced sharply”.
11. Line 343-345: This sentence needs to be supported by literature.
12. Line 356: FTIR and XPS are also part of biochar characterization, so it is not appropriate to put them into this part as a whole.
Figure: It is best to label the amount of adsorbent, solution pH, pollutant concentration and other experimental conditions.
Author Response
Response to Reviewers:
Reviewer #1:
The author studied and compared the adsorption mechanism of phosphate on three kinds of Fe-modified biochar. The research work is systematic and comprehensive. The research results are helpful to enhance the understanding of removal mechanisms of phosphate by Fe-modified biochar, and contribute to the application of Fe-modified biochar on phosphate removal in water. However, the mechanism of P adsorption by three kinds of Fe-modified biochar has not been well proved by experiments, so I think more experiments should be added to be more convincing.
Thank you for your valuable input and constructive advice. We have tried out best to improve the quality of the manuscript. We sincerely hope that this revised manuscript has addressed all your concerns and suggestion.
Introduction
- Line 40: change “the other” to “another”.
Thank you. We have changed it in Ln 41.
“Besides, runoff from agricultural lands that extensively applied P-containing fertilizers and pesticides is another dominant reason.”
- Line 71: It is inappropriate to use “besides” here, it should be a transition relationship.
Thank you. We have revised it in Ln 68.
“However, previous studies showed that Fe oxides enhance the removal of phosphate because of the large amounts of hydroxyl radicals on the surface as well as the Fe leaching can precipitate phosphate [19].”
- Line 83-85: The research significance needs to be rewritten. The focus of this paper should be to highlight the removal of phosphate.
Thank you. We have revised it in Ln 81-83.
“Our work can contribute to understanding better the removal mechanisms of phosphate by various Fe-modified biochar and its application.”
Materials and methods
- Line 103: “tried” ?, It should be “dried”.
Thank you. We have revised it in Ln 101.
“The solid residue was carefully collected and then dried at 80oC for 24 h for a consistent weight.”
Results and discussion
- Line 207: It is inappropriate to use “yet” here, it should be a causal relationship.
Thank you. We have revised it in Ln 207. We also checked the whole manuscript to replace it in Ln 363 and Ln 420.
“Among the four different biochars, ZBC has the highest pyrolytic temperature of 900oC, though it has the highest surface area (261.7 mg g-1) (Table 1), which is possibly associated with the ZVI itself.”
- Line 233: “starting from pH 3 for CSBC and pH 1 for GBC”. This sentence is contrary to the previous statement that the Fe-containing minerals can be dissolved under acidic conditions and needs to be rewritten.
Thank you. We checked the discussion again and amended these sentences in Ln 231-233.
“Interestingly, the ICP-OES results also suggested that the Fe-containing minerals on the surface of GBC and CSBC can be dissolved by acidic conditions, starting from different pH. For CSBC, is pH 3 and for GBC is pH 1.”
- Line 242: It is inappropriate to use “Besides” here, the conjunctions of the whole article need to be rechecked.
Thank you. We have revised it in Ln 244 and checked the whole manuscript.
“Nevertheless, the adsorption capacity of P obviously decreased with the increase of solution pH in the batch experiments.”
- Line 248-250: This sentence can be deleted or rewritten because the previous study has no obvious significance for the present study.
Thank you. We have removed the sentence.
- Line 265: Please explain why the Elovich model was not used to fit the adsorption kinetics.
Many thanks for your question. We used intra-particle diffusion model which is also called the Weber-Morris model because it is frequently used for phosphate removal, especially for porous adsorbents (Cui et al., 2020; Nardis et al., 2022; Peng et al., 2021). It is normally used to describe the adsorbate is diffused into the adsorbents’ pores. Our study showed well fit to the intra-particle diffusion model as well. The Elovich model is applied to describe adsorption on highly heterogeneous surface. We think it is good enough to include three different models for the adsorption kinetics.
Cui, Q., Xu, J., Wang, W., Tan, L., Cui, Y., Wang, T., Li, G., She, D., Zheng, J., 2020. Phosphorus recovery by core-shell γ-Al2O3/Fe3O4 biochar composite from aqueous phosphate solutions. Science of The Total Environment 729, 138892. https://doi.org/10.1016/j.scitotenv.2020.138892
Nardis, B.O., Franca, J.R., Carneiro, J.S. da S., Soares, J.R., Guilherme, L.R.G., Silva, C.A., Melo, L.C.A., 2022. Production of engineered-biochar under different pyrolysis conditions for phosphorus removal from aqueous solution. Science of The Total Environment 816, 151559. https://doi.org/10.1016/j.scitotenv.2021.151559
Peng, Y., Sun, Y., Fan, B., Zhang, S., Bolan, N.S., Chen, Q., Tsang, D.C.W., 2021. Fe/Al (hydr)oxides engineered biochar for reducing phosphorus leaching from a fertile calcareous soil. Journal of Cleaner Production 279, 123877. https://doi.org/10.1016/j.jclepro.2020.123877
- Line 266: Add “both” before "can".
Thank you. We have revised it in Ln 265.
“The adsorption of P by both ZBC and GBC can be divided into a fast and a slow adsorption stage, which quickly reached adsorption equilibrium in 120 and 240 min, respectively, compared with CSBC which has steady adsorption (Fig. 4a).”
- Line 280-282: There needs to be a phenomenon description, such as the lines did not cross through the origin.
Thank you. We have removed this discussion.
- Line 289-293: These two sentences need to be supported by literature, and the last sentence is too long to express clearly.
Thank you. We have revised this part in Ln 291-294.
“The adsorption results of CSBC, GBC, and ZBC fitted better with Freundlich model, owing to the higher R2 values of over 0.95, indicating the adsorption occurred more likely on the heterogeneous surface, which is tended to a multi-layer adsorption [40]. This also agreed with the results of kinetics studies that Fe-modified biochar fitted better with the pseudo-second-order double-layer adsorption.”
- Line 328-330: This sentence made no sense; I suggest deleting it.
Thank you for your suggestion. However, to have a better comparison with ZBC, we wish to keep this sentence.
“With the increase of desorption cycle times, CSBC and GBC showed low desorption efficiency (below 0.92%) in the 2nd and 3rd desorption cycle.”
- Line 341: change “sharply reduced” to “reduced sharply”.
Thank you. We have revised it in Ln 353.
“The results exhibited that though ZBC showed the highest removal efficiency of PO43− (96.52%) for the first washing, the capacity of PO43−removal by ZBC reduced sharply (24.98%) after desorption by NaOH (Fig. 9).”
- Line 343-345: This sentence needs to be supported by literature.
Thank you. We have revised it in Ln 355-357.
“This is strongly related to the ZVI aging effect and deactivation, and many previous studies have reported similar results [43,44].”
- Line 356: FTIR and XPS are also part of biochar characterization, so it is not appropriate to put them into this part as a whole.
Thank you for your suggestion. We agreed that they are biochar characterization. However, we put them here owing to the discussion order.
Figure: It is best to label the amount of adsorbent, solution pH, pollutant concentration and other experimental conditions
Thank you. We have revised it in the titles of Figures. Please see below.
“Figure 2. Effects of different pH on the P adsorption on BC, CSBC, GBC and ZBC. (Adsorbent dose: 5g L-1; P initial concentration: 15 mg L-1; adsorption time:4 h; temperature: 25 oC; ionic strength: 0.01 M)
Figure 3. Effects of different pH on the leaching amount of P on BC. (Adsorbent dose: 5g L-1; P initial concentration: 15 mg L-1; adsorption time:4 h; temperature: 25 oC; ionic strength: 0.01 M)
Figure 4. Adsorption kinetics of P by CSBC, GBC, and ZBC: (a) Pseudo-first-order and Pseudo-second-order and (b) Intra-particle diffusion model. (Adsorbent dose: 5g L-1; P initial concentration: 15 mg L-1; pH: 6; temperature: 25 oC; ionic strength: 0.01 M)
Figure 5. Adsorption isotherms of P by CSBC, GBC, and ZBC. (Adsorbent dose: 5g L-1; adsorption time: 4 h; pH: 6; temperature: 25 oC; ionic strength: 0.01 M)
Figure 6. Effects of different dosages on the P adsorption on CSBC, GBC, and ZBC. (P initial concentration: 15 mg L-1; adsorption time:4 h; pH: 6; temperature: 25 oC; ionic strength: 0.01 M)
Figure 7. Effects of ionic strength on the P adsorption on CSBC, GBC, and ZBC. (Adsorbent dose: 5g L-1; P initial concentration: 15 mg L-1; adsorption time: 4 h; pH: 6; temperature: 25 oC)
Figure 8. Effects of desorption on the P on CSBC, GBC, and ZBC. (Adsorbent dose: 5g L-1; P initial concentration: 15 mg L-1; adsorption time: 24 h; pH: 6; temperature: 25 oC; ionic strength: 0.01 M; desorption concentration: 0.1 M NaCl; desorption time: 24h)
Figure 9. The removal efficiencies of P at each reuse cycle by the CSBC, GBC and ZBC. (Adsorbent dose: 5g L-1; P initial concentration: 15 mg L-1; adsorption time: 24 h; pH: 6; temperature: 25 oC; ionic strength: 0.01 M; desorption concentration: 0.5 M NaOH; desorption time: 24h)”

Reviewer 2 Report
The manuscript entitled "Removal mechanisms of phosphate in aqueous solutions by three different Fe-modified biochars" from the authors Yiyin Qin, Xinyi Wu, Qiqi Huang, Jingzi Beiyuan, Jin Wang, Juan Liu, Wenbing Yuan, Chengrong Nie and Hailong Wang is very well written and presents the original research article. The subject of the paper is about the removal of the phosphate from the water media using the Fe-modified materials-zero- valent iron (ZVI), goethite and magnetite. The authors has presented in detail the modification of the biochar by the mentioned materials. The large number of the characterization technique is presented in detail.
In the present study, the results are well presented. The mechanism of the phosphorus removal with Fe-modified materials is well described and supported by fact.
Authors should only try to explain the low value of the R2 concerning the adsorption kinetic (page 8, row 272, Table 2).
According to the results of desorption, is the usage of this materials recommended for water media?
I recommend the acceptance in the present form, after the authors answer on the questions above.
Author Response
Reviewer #2:
The manuscript entitled "Removal mechanisms of phosphate in aqueous solutions by three different Fe-modified biochars" from the authors Yiyin Qin, Xinyi Wu, Qiqi Huang, Jingzi Beiyuan, Jin Wang, Juan Liu, Wenbing Yuan, Chengrong Nie and Hailong Wang is very well written and presents the original research article. The subject of the paper is about the removal of the phosphate from the water media using the Fe-modified materials-zero- valent iron (ZVI), goethite and magnetite. The authors has presented in detail the modification of the biochar by the mentioned materials. The large number of the characterization technique is presented in detail. In the present study, the results are well presented. The mechanism of the phosphorus removal with Fe-modified materials is well described and supported by fact.
We are grateful to the learned Reviewer for praising our effort.
Authors should only try to explain the low value of the R2 concerning the adsorption kinetic (page 8, row 272, Table 2).
Thank you for your suggestion. The possible low R2 for the adsorption kinetic might be caused by the mechanisms are complicated for the modified biochar. The mechanisms cannot be simply explained by physical and chemical sorption in our study considering the relatively low R2. The fitting is better for the intra-particle diffusion model, however, the R2 was missing in Table 2 our previously submitted version. Now we have included the R2 of the intra-particle diffusion model. We also amended our manuscript in Ln 282-288 as below.
“The three-stage intra-particle diffusion model was used to further analyze the adsorption processes of P on the Fe-modified biochars (Fig. 4b). GBC and ZBC have slippery slopes in the first stage (within 1 h), indicating that the GBC and ZBC rapidly removed PO43- by the adsorption sites provided by Fe ions in the first stage, then the adsorption sites were gradually occupied. Remarkably, the easily accessible sorption sites of ZBC have been quickly occupied and the adsorption reduced distinctly. The adsorption by CSBC showed a significant difference from ZBC and GBC, a slow adsorption within the first 1 h and a long and flat adsorption in the second and third stages were observed.”
Table 2. Parameters of adsorption kinetics for P removal by GBC, CSBC, and ZBC.
|
Pseudo-first order |
Pseudo-second order |
|
Intra-particle diffusion |
||||||||||||||
|
Biochar |
Qe |
K1 |
R2 |
Qe |
K2 |
R2 |
|
C1 |
K3 |
R2 |
C2 |
K4 |
R2 |
C3 |
K5 |
R2 |
|
|
CSBC |
0.935 |
0.003 |
0.587 |
|
1.223 |
0.006 |
0.685 |
|
0.307 |
0.011 |
0.999 |
0.080 |
0.030 |
0.994 |
0.704 |
0.012 |
0.463 |
|
GBC |
1.783 |
0.018 |
0.607 |
|
1.90 |
0.022 |
0.796 |
|
0.645 |
0.104 |
0.995 |
1.406 |
0.016 |
0.981 |
1.753 |
0.006 |
0.455 |
|
ZBC |
2.794 |
0.087 |
0.677 |
|
2.819 |
0.136 |
0.715 |
|
1.335 |
0.190 |
0.991 |
2.806 |
-7.082 |
0.596 |
2.961 |
-0.005 |
0.980 |
According to the results of desorption, is the usage of this materials recommended for water media?
Many thanks for your question. The desorption of ZBC is high, compared with GBC and CSBC, though it has a higher removal capacity. GBC is more recommended for serving as adsorption media, such as in permeable reactive barriers to remove phosphate in waterbody. Maybe ZBC can be applied in some quick action for P adsorption, owing to its high Qm and quick adsorption.
I recommend the acceptance in the present form, after the authors answer on the questions above.
Much appreciated for your positive feedbacks again.

Reviewer 3 Report
In this study authors evaluated the adsorption mechanisms of phosphate on the three Fe-modified biochar. The study is very comprehensive as far as the experiments are concerned, the results are clearly and thoroughly explained. Certain deficiencies just need to be corrected before the manuscript is published.
1. In part 2.1. please add information about the used pyrolysis furnace.
2. Is there reference available for preparation of ZVI-modified biochar?
3. What technique was used to determine the content of the elements in table 1?
4. Is it possible to support the statement in lines 240-241 with literature?
5. It is necessary to compare the obtained results of the adsorption study with the literature.
Author Response
Reviewer #3:
In this study authors evaluated the adsorption mechanisms of phosphate on the three Fe-modified biochar. The study is very comprehensive as far as the experiments are concerned, the results are clearly and thoroughly explained. Certain deficiencies just need to be corrected before the manuscript is published.
Thank you very much your meticulous review and valuable suggestion.
- In part 2.1. please add information about the used pyrolysis furnace.
Thank you. We have revised it in Ln 94.
“The pristine biochar (BC) was pyrolyzed at 600oC at a heating rate of 5oC min-1 with flushing of N2 (0.3 L min-1) for 1 h in a tube furnace (Shengli SLG1100, China).”
- Is there reference available for preparation of ZVI-modified biochar?
We slightly amended the method as below. We marginally revised the preparation of ZVI-modified biochar in the reference and used in our study in Ln 111.
“ZVI-modified biochar (ZBC) was produced by mixing 1.93 g FeCl3∙6H2O and 20 g feedstock with 800 mL DIW for 24 h, then heating at 80oC in a water bath for 8 h, which is adapted from Han et al. [25].”
- What technique was used to determine the content of the elements in table 1?
Thank you. An elemental analyzer was used. We have revised it in Ln 124-125.
“An elemental analyzer (Elementar UNICUBE, Germany) was used to determine the total content of C/H/N/O/S of the biochars.”
- Is it possible to support the statement in lines 240-241 with literature?
Thank you. We have revised it in Ln 242-244.
“This suggested that the Fe-containing minerals of GBC might be the most important components for PO43− adsorption in the current study [36].”
- It is necessary to compare the obtained results of the adsorption study with the literature.
Many thanks for your nice suggestion. We revised the manuscript and included a Table S2 to compare the adsorption capacity with previous studies. Please see in Ln 305-311 and below.
“Though the adsorption capacity of the Fe-modified biochar is relatively low compared with some adsorbents (Table S2), for example, biochar modified by Mg or Ca [16,41], it is significantly improved compared with the pristine biochar. Compared with some Fe-modified biochars, such as Fe-impregnated biochars derived from wood chip (Qm=3.201 mg g-1) and magnetic biochar derived from water hyacinth (Qm=5.07 mg g-1), GBC and ZBC in the current study have better adsorption effects.”

Round 2
Reviewer 1 Report
The authors have revised the suggestion according to the reviewers.
I think it should be accepted now.